# Macro-scale relationship between body mass and timing of bird migration

Xiaodan Wang [1,2], Marius Somveille[2], Adriaan M. Dokter [3], Wenhua Cao[1], Chuyu Cheng[1], Jiajia Liu [1] & Zhijun Ma [1] ✉

Clarifying migration timing and its link with underlying drivers is fundamental to understanding the evolution of bird migration. However, previous studies have focused mainly on environmental drivers such as the latitudes of seasonal distributions and migration distance, while the effect of intrinsic biological traits remains unclear. Here, we compile a global dataset on the annual cycle of migratory birds obtained by tracking 1531 individuals and 177 populations from 186 species, and investigate how body mass, a key intrinsic biological trait, influenced timings of the annual cycle using Bayesian structural equation models. We find that body mass has a strong direct effect on departure date from non-breeding and breeding sites, and indirect effects on arrival date at breeding and non-breeding sites, mainly through its effects on migration distance and a carry-over effect. Our results suggest that environmental factors strongly affect the timing of spring migration, while body mass affects the timing of both spring and autumn migration. Our study provides a new foundation for future research on the causes of species distribution and movement.

Every year, billions of migratory birds fly thousands or even tens of thousands of kilometres between breeding and non-breeding sites with predictable timing. Bird migration is not only a fascinating natural phenomenon, but also an important global ecological process that involves interaction with local biological communities and promotes the flow of materials, energy, and information along migration routes[1]. The annual cycle of migratory birds is composed of a series of events and biological activities that are strictly timed, generally by internal biological clock[2], to match the periodical changes in the environment in order to maximize fitness[3]. Clarifying migration timing and its link with underlying drivers is fundamental to understanding the development, maintenance, and evolution of bird migration[4,5] as well as to predict the responses of migratory birds to global environmental changes[6,7].

Seasonal changes in environmental conditions are the major external driving force for migration[1,8]. Within this context, latitude, which is strongly associated with local temperature and phenology, is a key factor that affects the migration schedule of birds[9]. Because suitable breeding times occur later at higher latitudes, birds that breed at higher latitudes generally depart from non-breeding sites and arrive at breeding sites later than those that breed at lower latitudes[10–12]. The latitude of the non-breeding sites can also affect migration timing[13,14]. Birds wintering closer to the equator, where environmental conditions are relatively stable and thus birds sense late external signals to migrate, tend to start their spring migration and arrive at their breeding sites later than those wintering at higher latitudes[14,15]. In addition, the geographic locations of breeding and non-breeding sites determine migration distance, which, in turn, affects migration duration[16] and, ultimately, migration timing[17]. With increasing distance, birds spend longer on migratory flight and require more refuelling, thus causing an earlier departure and/or later arrival.

[1]Shanghai Institute of Infectious Disease and Biosecurity, Ministry of Education Key Laboratory for Biodiversity Science and Ecological Engineering, National Observations and Research Station for Wetland Ecosystems of the Yangtze Estuary and Institute of Eco-Chongming, School of Life Sciences, Fudan University, Shanghai 200438, China. [2]Centre for Biodiversity and Environment Research, Department of Genetics, Evolution and Environment, University College London, London WC1E 6BT, UK. [3]Cornell Lab of Ornithology, Cornell University, Ithaca, NY 14850, USA. ✉e-mail: zhijunm@fudan.edu.cn

Migration timing can also be influenced by intrinsic biological traits. In particular, body mass affects the physiology, behaviour, and ecology of organisms[18–20] and, thus, is closely associated with migration timing as it can affect the amount of time required for various activities across the annual cycle. An analysis of the scaling of the annual cycle in birds suggests that larger species spend longer periods breeding one brood, which is largely driven by longer incubation periods due to larger eggs[21], and longer time to grow into adult size[22]. It has also been shown that migration speed (i.e., migration distance divided by total travel time) decreases with increasing body mass during powered flapping flights[23–25]. Thus, the migration of larger species is expected to be subject to stronger time constraints than that of smaller species[26]. Because breeding and migration are all time-consuming activities, with increasing time investment with body mass[21], we expect that to maximize fitness larger species exhibit earlier migration timing in spring, which has been observed in some shorebird species[27]. Although the duration of the breeding period per brood for larger species is generally longer, smaller species may produce multiple broods during the breeding season when the timing is suitable. Therefore, there might be no consistent effect of body mass on the departure date of migratory birds from their breeding sites[27].

Moreover, body mass affects the latitudinal distribution of species[28], and thus can in turn indirectly affect migration timing. Larger species have a better tolerance to cold than smaller species[26] and therefore they can live in colder areas at higher latitudes, following Bergmann's rule[26]. However, breeding seasons are relatively short at high latitudes, which may limit the ability of larger species to breed there as they tend to have a higher time requirement for the breeding period[21]. We therefore predict that larger species tend to breed at lower latitudes than smaller species, but that they spend the non-breeding season at relatively high latitudes compared to smaller species due to their higher tolerance to colder climate, which would result in a shorter migration distance and migration period.

Although it is widely recognized that bird migration is regulated by both external and internal factors[3,16,29], most previous studies on migration timing focused on the influence of environmental factors and on a limited number of species and flyways[12,30]. Only a few papers discussed the relationship between body mass and migration timing and they only addressed one-way migration period in a few bird groups[27,30]. For example, Zhao et al. found that departure date from the non-breeding sites and arrival date at the breeding sites were negatively related to body mass based on seven species in family Scolopacidae that wintered in Oceania[27]. According to radar observation in the northeastern United States, La Sorte et al. indicated that small-sized nocturnal migrants tended to depart from breeding sites earlier[30]. A comprehensive understanding of the global spatiotemporal dynamics of bird migration and its determinants, including both intrinsic biological traits and external drivers, is still lacking.

Due to the rapid development of tracking technology, the annual schedules of many migratory birds have been accurately described. These data now make it possible for a comprehensive analysis of migration timing and related factors to be conducted in multiple species on a global scale. To fully understand the impacts of intrinsic biological traits and external drivers on the migration timing of birds, we compiled available data on the full annual cycles of 1531 individuals and 177 populations of 186 bird species of 44 families of 19 orders that breed in the Northern Hemisphere from 306 published tracking studies (Fig. 1, Supplementary Fig. 1, Supplementary Table 1 and Supplementary Data 1). Using Bayesian phylogenetic structural equation models (SEMs), we analyzed the effects of environmental factors (latitude of breeding and non-breeding sites and migration distance) and body mass on the timing of the four key events of the annual cycle of migratory birds: departure from non-breeding sites and arrival at breeding sites during spring migration, and departure from breeding sites and arrival at non-breeding sites during autumn migration.

We tested the following four hypotheses: (1) body mass of migratory birds affects the latitudinal distribution of their breeding and non-breeding sites, as well as their migration distance; with larger species tending to breed at low latitudes but wintering at relatively high latitudes, and thus exhibiting shorter migration distances than smaller species; (2) body mass affects migration timing, with larger species exhibiting earlier migration timing in spring; (3) latitudes of breeding and non-breeding sites affect migration timing, with low-latitude breeding sites, high-latitude non-breeding sites, related to earlier migration timing; and (4) migration distance affects migration timing, with shorter migrations related to later migration departure and earlier arrival.

In this work, we investigate the key factors that affect the migration timing of birds. We demonstrate that environmental factors (breeding and non-breeding latitude and migration distance) strongly affect the timing of spring migration, while intrinsic biological trait (body mass) affects the timing of both spring and autumn migration. Our findings highlight the impacts of size-related traits on the spatiotemporal patterns of migratory birds.

## Results

### Time allocation of migratory birds throughout the annual cycle

Among the 186 species included in this study, the mean dates of departure from non-breeding site, arrival at breeding site, departure from breeding site, and arrival at non-breeding site were 27 March (SD, ±24 days), 2 May (± 27 days), 26 August (± 35 days), and 16 October (± 35 days), respectively (Supplementary Fig. 2). The non-breeding period was the longest in duration (mean ± SD, $163 \pm 46$ days, 44.5% of the total cycle), followed by the breeding period ($116 \pm 47$ days, 31.8%). The total migration period, including both spring and autumn migration, accounted for 23.6% of the annual cycle. The autumn migration period ($51 \pm 36$ days) was generally longer than the spring migration period ($35 \pm 23$ days, paired $t$ test, $t = 29.02$, $df = 1707$, $P < 0.001$, Supplementary Fig. 2).

Bayesian linear mixed models indicated that body mass was significantly associated with the length of the breeding and non-breeding periods, with larger birds staying longer on breeding sites (95% credible interval (CI) [0.20, 0.58]) (Supplementary Fig. 3b and 4b) and a shorter amount of time on non-breeding sites (95% CI [−0.63, −0.18]) (Supplementary Fig. 3d and 4d). However, body mass did not significantly impact the duration of the spring (95% CI [−0.17, 0.26]) (Supplementary Fig. 3a and 4a) or autumn (95% CI [−0.15, 0.16]) migration periods (Supplementary Fig. 3c and 4c).

### Effect of body mass on latitudinal distribution

SEM analyses demonstrated a significant correlation between body mass and non-breeding latitudes of the tracked birds, with larger species tending to winter at higher latitudes (95% CI [0.33, 0.78] with full dataset, Figs. 2 and 3; 95% CI [0.32, 0.79] with data of birds captured at breeding sites, Supplementary Fig. 5b). However, body mass did not exhibit a significant latitudinal gradient in the breeding season (95% CI [−0.12, 0.35] with full dataset, Fig. 2 and Supplementary Fig. 6a; 95% CI [−0.54, 0.23] with data of birds captured at non-breeding and stopover sites, Supplementary Figs. 5a and 6b). SEM analyses also showed that migration distance was negatively related to body mass (95% CI [−0.57, −0.20], Fig. 2 and Supplementary Fig. 7c). We found that 34 of the total 186 species in this study had an average latitude of non-breeding sites in south of the equator. However, the average latitudes of non-breeding sites for each species weighing more than 1.1 kg (36 species, including some geese, cranes, storks, and eagles, etc.) were all located north of the equator (Supplementary Fig. 6c).

### Effect of body mass on migration timing

Body mass showed strong direct effects on the departure date from non-breeding and breeding sites, with larger species tending to start

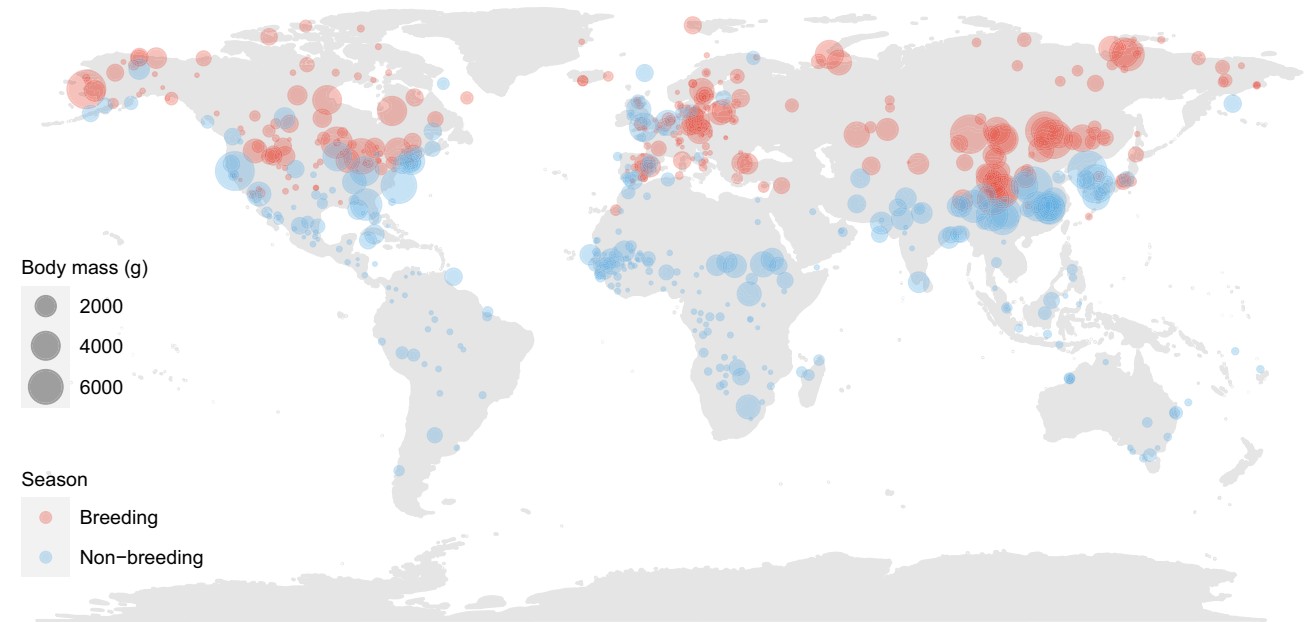

**Fig. 1 | Distribution of the breeding (red dots) and non-breeding (blue dots) sites of birds included in this study.** The figure shows data from 1531 individuals and 177 populations, obtained from 306 studies of 186 species. Each dot represents the median longitude and latitude for each species in each study.

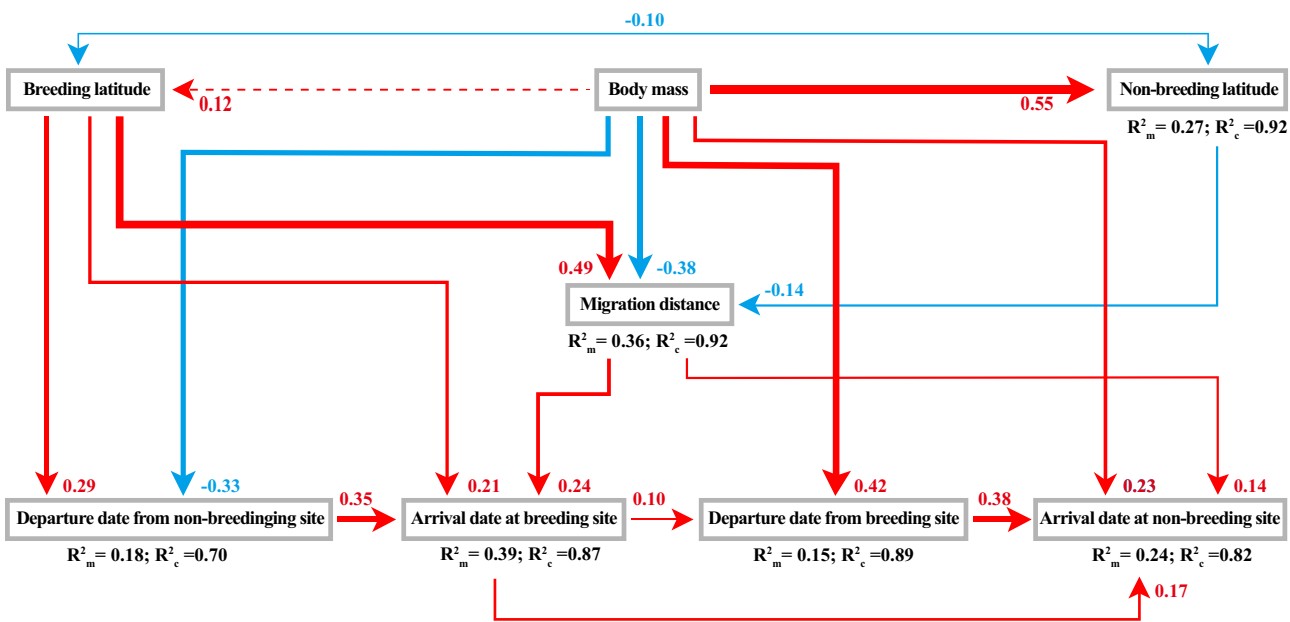

**Fig. 2 | Results of the structural equation model analysis on migration timing of 186 species.** Arrows represent the direct effects of variables on the migration timing of birds. Red arrows and values (correlation coefficients) represent positive significant effects (credible intervals excluding zero); blue arrows and values (correlation coefficients) represent negative significant effects (credible intervals excluding zero). The dashed-line arrows represent non-significant relationships (credible intervals including zero). $R^2_m$, marginal $R$ square, representing only the variance of the fixed effects, $R^2_c$, conditional R square, representing both the fixed and random effects. The absolute value of the non-breeding latitude was used as the non-breeding latitude.

migration earlier in spring and later in autumn (Figs. 2 and 3; Table 1). Moreover, body mass had indirect and negative effects on the arrival date at the breeding site, mediated by migration distance, non-breeding latitude and the date of departure from the non-breeding site (Fig. 2; Table 1). Additionally, body mass directly affected the arrival date at the non-breeding site and exhibited indirect effects mediated by migration timing in earlier periods (positive effect) and migration distance (negative effect), with larger species tending to end their autumn migration later (Fig. 2, Supplementary Fig. 8; Table 1).

Further analysis on Passeriformes (53 tracked species) indicated that breeding latitude but not body mass significantly affected annual migration timing (Supplementary Fig. 9).

### Effect of geographical distribution on migration timing

SEM analyses showed that higher-latitude breeders generally have later migration timings than lower-latitude breeders throughout the entire annual cycle (Figs. 2 and 4a, b; Table 1). We also found that breeding latitude directly influenced migration timing in spring but indirectly influenced migration timing in autumn (Fig. 2; Table 1), the

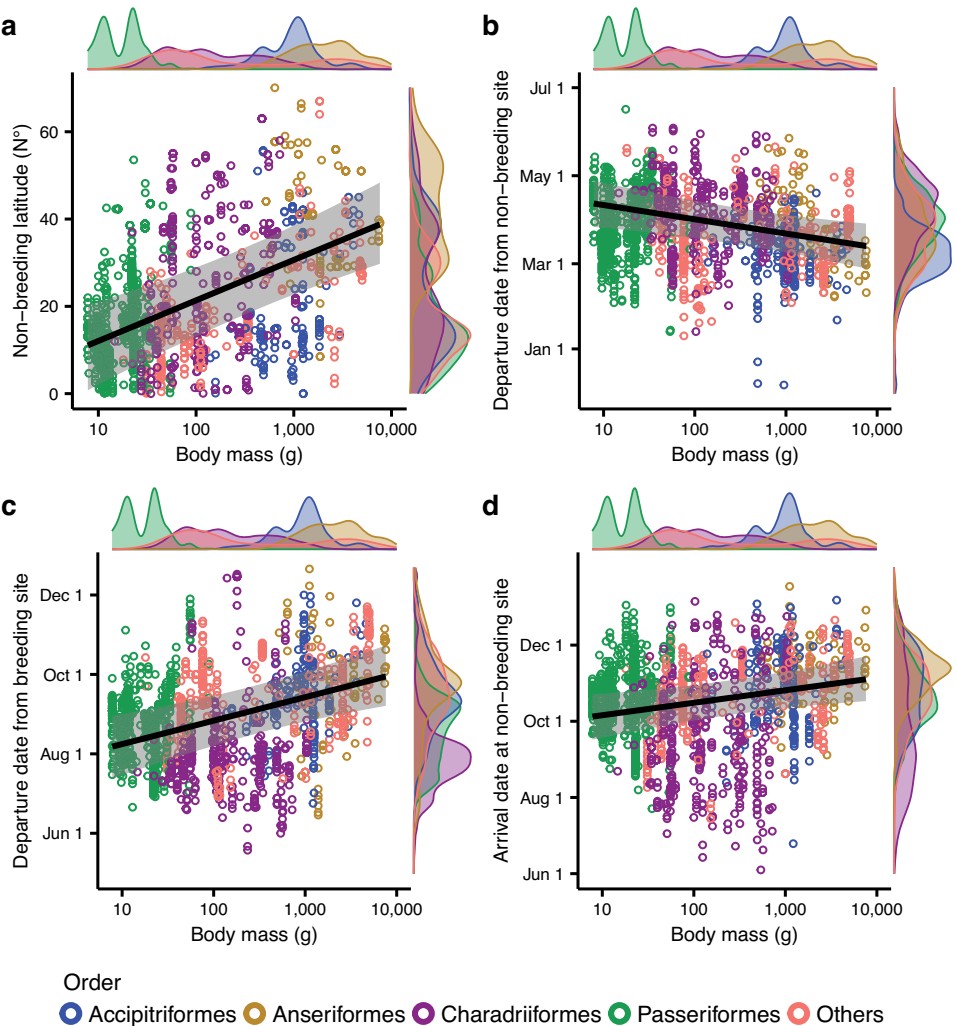

**Fig. 3 | Direct effect of body mass on spatiotemporal migration patterns of birds in structural equation models.** Direct effects of body mass on the non-breeding latitude (**a**), where absolute values were used for birds wintering in the Southern Hemisphere, departure date from the non-breeding site (**b**), departure date from the breeding site (**c**), and arrival date at non-breeding site (**d**) of migratory birds. Light bands represent 95% credible intervals. Dots represent each tracked individual or population. The density distribution of each variable is plotted alongside the scatter plots.

latter being attributed to migration distance and carry-over effects of timing in earlier periods (Fig. 2). Non-breeding latitude also exerted indirect effects on the timing of arrival at breeding sites and non-breeding sites during migration by affecting migration distance (Fig. 2; Table 1). Migration distance exerted a direct effect on arrival dates in both spring and autumn migration: the longer the migration distance, the longer the migration periods and the later the date of arrival at the migration destination (Figs. 2, 4c, d and Supplementary Fig. 3a, c). However, migration distance had no significant impact on the departure dates from non-breeding or breeding sites (Fig. 2; Table 1).

**Effects of other variables on migration timing**
Carry-over effects affect migration timing within an annual cycle. In particular, departure date strongly impacts arrival date in both spring and autumn migration (Fig. 2, Supplementary Fig. 8). In addition, the SEM model variance explained by phylogenetic relationship and year of tracking was small compared with intraspecific variance or paper ID and species ID (Supplementary Fig. 10). Flight mode (soaring or flapping) had no significant impact on migration distance and any timing of the four key events of the annual cycle (Supplementary Fig. 11).

When sex was included in the models, it was observed to directly impact the arrival date at the breeding site, with males arriving at breeding sites earlier than females (Supplementary Figs. 12 and 13). The impacts of other variables on migration timing were consistent with earlier descriptions when sex was not included in the models (Supplementary Fig. 12 and Supplementary Table 2), suggesting the results are robust.

## Discussion
Using the most extensive dataset on bird migration timing examined to date, our results indicate that body mass is an important determinant of bird migration, affecting migration timing directly but also indirectly through influencing latitudinal distribution and migration distance. Our findings highlight that size-related capabilities and constraints strongly influence the spatiotemporal patterns of bird migration across the full annual cycle.

In support to our first hypothesis, we found that body mass does not exhibit a significant latitudinal gradient in migratory birds during their breeding season, according to data of all the birds or data of birds captured outside the breeding sites. This result contrasts with the pattern observed in sedentary birds[31] and other animal groups such as mammals[32], fish[33], and marine copepods[34]. This may be because temperatures remain generally mild to warm across the Northern Hemisphere during the breeding season (i.e., northern summer), and

**Table 1 | Standardized direct effects, indirect effects, and total effects between four variables and four migration timings among migrant species estimated by structural equation model analysis**

| Timing | Explanatory variable | | | | | | | |
|---|---|---|---|---|---|---|---|---|
| | Effect | Body mass | Breeding latitude | Non-breeding latitude | Migration distance | Departure date from non-breeding site | Arrival date at breeding site | Departure date from breeding site |
| Departure date from non-breeding site | Direct | −0.33 | 0.29 | - | - | - | - | - |
| | Indirect | - | - | - | - | - | - | - |
| | Total | −0.33 | 0.29 | - | - | - | - | - |
| Arrival date at breeding site | Direct | - | 0.21 | - | 0.24 | 0.35 | - | - |
| | Indirect | −0.23 | 0.22 | −0.03 | - | - | - | - |
| | Total | −0.23 | 0.43 | −0.03 | 0.24 | 0.35 | - | - |
| Departure date from breeding site | Direct | 0.42 | - | - | - | - | 0.10 | - |
| | Indirect | −0.02 | 0.04 | −0.00 | 0.02 | 0.04 | - | - |
| | Total | 0.40 | 0.04 | −0.00 | 0.02 | 0.04 | 0.10 | - |
| Arrival date at non-breeding site | Direct | 0.23 | - | - | 0.14 | - | 0.17 | 0.38 |
| | Indirect | 0.05 | 0.16 | −0.03 | 0.05 | 0.07 | 0.04 | - |
| | Total | 0.28 | 0.16 | −0.03 | 0.19 | 0.07 | 0.21 | 0.38 |

Non-breeding latitude is the absolute value for birds wintering in the Southern Hemisphere. "-" means no significant effect.

migratory birds can avoid cold winter temperatures. Therefore, larger body mass might not be required for adaptation to colder climates at breeding sites in migratory birds. Additionally, the suitable breeding period is short at high latitudes, which limits the breeding of large species that require long breeding periods[10]. To mitigate the time constraint of a short breeding period at high latitudes, some large species, such as geese and swans, adopt a capital breeding strategy and accumulate fuel and nutrients from stopover sites to support breeding activities[35], thus the time investment at breeding sites is reduced and their chicks can match the seasonal food peak at breeding sites. Such adaptation might explain why we do not see an inverse relationship between body mass and breeding latitude in migratory birds.

During the non-breeding season, we found that the distribution of migratory birds is consistent with Bergmann's rule as larger species tend to winter at higher latitudes. This result supports our first hypothesis, and it could be due to the higher tolerance of large birds to cold[26,28]. Our results also indicate that larger birds tend to migrate shorter distances than smaller birds, which is consistent with their higher time requirements for breeding[36]. Consequently, long-distance migratory birds, especially those wintering in the Southern Hemisphere, tend to be smaller species (Fig. 2, Supplementary Fig. 6c), which have relatively lower time and energy requirements for breeding per brood compared with larger species[21,36], potentially allowing them to invest more time and energy in migration activities. This result contrasts with a previous study on the migration of seven species in family Scolopacidae, which found that body mass did not affect migration distance[27]. However these previous results were likely due to the fact that all the species in the study were relatively small-sized species (body mass from 50 to 750 g) with concentrated non-breeding sites in Oceania[27], thus making the relationship undetectable.

In support to our second hypothesis, we found that larger birds spent longer time at the breeding sites than smaller birds, largely driven by a direct effect of body mass on the departure date from breeding sites, i.e., larger birds start their autumn migration later than smaller birds. We found that body mass also affects the arrival date at breeding sites but this is an indirect effect mediated by migration distance, i.e., larger birds have shorter migration distance which leads to earlier arrival on breeding sites. The smaller, indirect effect of body mass on arrival date at breeding sites is likely affected by constraints due to the prevalence of harsh environmental conditions if they arrive too early[1]. The longer duration in the breeding sites of larger birds was compensated by a shorter duration in the non-breeding site, while migration duration is not related to body mass during either spring or

autumn despite larger species migrating shorter distances, which is consistent with that the migration speed of large species is generally slower than that of small species[23–25]. These results suggest that large species increase their investment in time and energy for breeding by reducing the time spent at non-breeding sites. We found that the migration timing of Passeriformes was strongly affected by breeding latitude but not body mass (Supplementary Fig. 9). This might be due to the smaller body mass of Passeriformes (<90 g for the 53 species in this study) makes them sensitive to periodical environmental changes, and thus annual migration timing is endogenously controlled[12]. This also suggests the impacts of body mass on migration timing mainly occur among taxonomic groups.

Our results show that birds breeding at higher latitude generally have later migration timing throughout the annual cycle, which supports our third hypothesis and is consistent with previous intraspecific studies that found that breeding latitude is the main driver of migration timing[10,12,37,38]. More specifically, our results indicate that latitude of breeding sites directly and positively influences migration timing in spring, i.e., birds breeding at higher latitude migrate later in the spring, but its influence on migration timing in autumn is indirect and mediated by migration distance and carry-over effects.

We found that migration distance also has a direct effect on timing, with longer-distance migrants terminating their migration later than shorter-distance migrants, which is consistent with our fourth hypothesis. However, in contrast with our hypotheses, non-breeding latitude did not affect migration timing directly and migration distance did not affect departure dates. This could be partially due to effects going in opposite directions cancelling each other: birds wintering at low latitude are expected to depart late but they also tend to migrate longer distances, which is expected to lead to earlier departure, thus potentially explaining why we did not find any effect.

This study illustrates how environmental factors (latitude and distance), intrinsic biological traits (body mass), and their interaction can influence key timings in the annual cycle of migratory birds. In particular, our results suggest that external factors strongly constrain the timing of spring migration as migratory birds must time their arrival on their breeding sites to match favourable environmental conditions and peak in food supply, while body mass affects the timing of both spring and autumn migration as larger birds require more time to breed. In response to global warming, many migratory birds advance migration timing in spring. In addition, some species are showing shrinking body sizes[39,40], shifts in their breeding and non-breeding ranges[41,42] and corresponding changes in migration

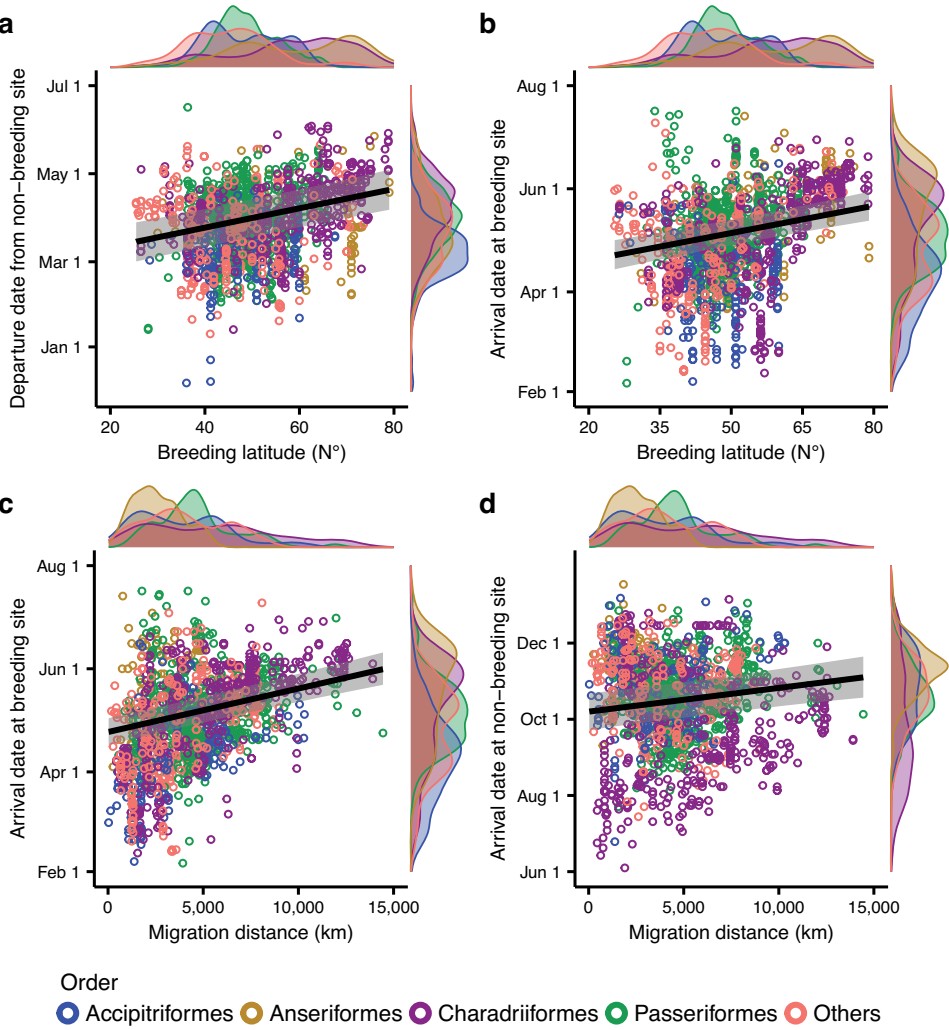

**Fig. 4 | Relationships of the timings of key events in migratory birds with environmental drivers in structural equation models.** The relationships between migration timings in spring and breeding latitudes (**a**, **b**) and the relationships between arrival dates and migration distances (**c**, **d**). Light bands represent 95% credible intervals. Dots represent each tracked individual or population. The density distribution of each variable is plotted alongside the scatter plots.

distances. Our results suggest that all these changes can also affect migration timing both directly and indirectly, and this work therefore provides a reference for further studies investigating how the migration timing of migratory animals is adjusted to adapt to climate change.

## Methods

### Literature search
We conducted a search for all papers that were published between 1 January 1900 and 1 January 2022 and that described the migration timing of individual birds. In the Web of Science (All Databases) we used the search terms "TS = (GPS OR PTT OR satellite OR geolocat*) AND (bird* OR aves OR avian) AND migrat*) AND PY= (1900-2021)", and in Scopus we used the search terms "TITLE-ABS KEY (gps OR ptt OR satellite OR geolocat*) AND TITLE-ABS KEY (bird* OR aves OR avian) AND TITLE-ABS-KEY (migrat*)) AND PUBYEAR < 2022". A total of 2706 papers were retrieved after duplicates were removed. We examined the title and abstract of each paper to exclude irrelevant studies (2194), such as studies on habitat use, local movements, home ranges or the development of transmitters. We then read the full text of each remaining paper and extracted the studies that tracked bird migration for at least one year (including both northward and southward migration) with complete data of all four key events of the annual cycle

(departure from non-breeding sites, arrival at breeding sites, departure from breeding sites, and arrival at non-breeding sites). If multiple papers used the same data sources, only the earliest published paper was included. We collected an additional 30 papers by checking the references of the searched papers. Finally, a total of 306 papers were included in our database. A flowchart of the selection process is shown in Supplementary Fig. 1.

### Data collection
We extracted the following complete data from each selected paper: species and/or subspecies name, individual identification, geographical coordinates of breeding and non-breeding sites, departure date from non-breeding site, arrival date at breeding site, departure date from breeding site, arrival date at non-breeding site, capture site and year when tracking was conducted. We also recorded the sex of individuals if the information was provided. Because birds of different ages show differences in migration timing[43-45], we only collected data for adult birds, excluding data for juveniles and subadults. We excluded birds breeding in the Southern Hemisphere because they have different migration patterns from those breeding in the Northern Hemisphere. Birds breeding in the tropics as well as seabirds (e.g., penguins, petrels, pelagic gulls and terns, and auks) were also excluded from the database because their migration behaviour often lacks

regular seasonal variation[46]. However, we included four species of gulls and three species of terns that often move in inland and coastal regions (Supplementary Data 1). If the data were depicted only in figures in the original studies, we used GetData Graph Digitizer (V2.26, www.getdata-graph-digitizer.com) to obtain original values, including the migration schedules and geographical coordinates of breeding and non-breeding sites. Some studies reported migration timing only at the population level and not at the individual level; for those, we extracted population-level data to represent a proxy for individual-level data. We generalized both individual-level data (1531) and population-level data (177) as "tracking data" in the description of this paper. Overall, in our analysis, we used 1708 full annual tracking data of 186 species in 44 families and 19 orders (Supplementary Table 1 and Supplementary Data 1). The 186 species were 12.0% of the total 1550 migratory species breeding in the Northern Hemisphere[47].

Some individuals stayed at more than one non-breeding site within one season. For those individuals, we used the first non-breeding site they visited and the date of arrival at that site as the non-breeding site and the arrival date for the autumn migration, respectively, and we used the last non-breeding site they visited and the date of departure from that site as the non-breeding site and the departure date from the non-breeding site in the spring migration. The period between arrival at the first non-breeding site and departure from the last non-breeding site was used as the non-breeding period. We defined the distance between the breeding site and the first non-breeding site as the migration distance in autumn and the distance between the last non-breeding site and the breeding site as the migration distance in spring. We calculated the great circle distance between the breeding and non-breeding sites as the migration distance for each individual using the *distm* function in the *Geosphere* package in R[48]. For the studies that reported migration and distribution data only at the population level, we used the mean data at the population level.

The body mass of migratory birds varies throughout the year, especially in long-distance migratory species, due to the large amount of fuel deposition and consumption in the migration season[49]. We used the minimum body masses, which are equivalent to the lean body mass, of females and males at the species or subspecies level, obtained from[50], as the body mass of each individual in this study. If sex was not reported in the study, we used the average of the minimum body masses of both sexes. The body mass of the bird species included in this paper ranged from 8.7 g to 7600 g, covering most of the body mass range of all migratory bird species in the world (Supplementary Fig. 14).

## Data analysis

To detect the time allocation of migratory birds throughout the annual cycle, we combined both tracking data reported at the individual level (1531) and population average (177) of a total of 186 species as "tracking data" (1708) and calculated the mean migration timing for each species and then the mean across species of the four key events of the annual cycle (departure from non-breeding sites, arrival at breeding sites, departure from breeding sites, and arrival at non-breeding sites). We defined the period between departure from the non-breeding sites and arrival at the breeding sites as the spring migration period, the period between arrival at the breeding sites and departure from the breeding sites as the breeding period, the period between departure from the breeding sites and arrival at the non-breeding sites as the autumn migration period, and the period between arrival at the non-breeding site and departure from the non-breeding sites as the non-breeding period.

To test the effects of geographical factors (latitude of breeding and non-breeding sites and migration distance) and body mass on the timing of the four key events of the annual cycle of migratory birds, we performed Bayesian phylogenetic piecewise structural equation

modelling (SEM)[51] on individual tracking data using the *brms* package[52,53]. The SEM was constructed according to the hypothesized relationships among the variables below:

(1) The latitudes of breeding and non-breeding sites determine the phenology and the migration distance, which is closely related to migration duration, and thus directly[10] and indirectly (via migration distance)[54] influences the timing of the start and end of both northward and southward migration.

(2) Body mass is related to the latitude of breeding and non-breeding sites following Bergmann's rule[26]; body mass also impacts the cost of migration[24]. Consequently, body mass can directly and indirectly affect migration distance[26,55] and the timing of the start and end of migration[27].

(3) The timing at earlier phases of the annual cycle influences the timing at later phases[14], so the migration timing at the earlier phases of the same migration cycle was included as an explanatory variable in the models. Birds can reset migration timing during the non-breeding period[56], so the migration timing at the earlier annual cycle has no carry-over effects on the migration timing at the following annual cycle.

Because the latitude of the non-breeding site had a strong impact on migration distance (95% CI [−0.89, −0.84]) (the farther south the non-breeding site was, the longer the migration distance was) (Supplementary Fig. 15), the absolute value of the latitudes of non-breeding sites in Southern Hemisphere was used in the SEM to avoid the issue of multicollinearity. To address the issue of nonrandom sampling in analyzing the relationship between body mass and breeding/non-breeding latitudes, we further assessed the relationship between body mass and breeding latitude by using the combined data of birds captured at stopover and non-breeding sites. We also assessed the relationship between body mass and non-breeding latitude by using the data of birds captured at breeding sites. Considering that flight mode is related with energetic consumption during flight and thus can affect migration ranges and migration timing[57], we classified flight mode into soaring and flapping[24] and included flight mode in the SEMs.

To control for the potential effects of phylogenetic relatedness on migration timing, phylogenetic relationship among species was included in the models as a random effect. We built a maximum clade credibility tree derived from the phylogenetic tree distributions[58]. We pruned the global phylogenetic tree of birds from BirdTree.org using the option 'Hackett All Species: a set of 10000 trees with 9993 OTUs each' to download bird species datasets[59]. We randomly sampled 1000 pseudo-posterior distributions and then constructed the maximum clade credibility tree using common ancestor heights with TreeAnnotator software from the BEAST package[60,61]. We calculated phylogenetic similarity at the species level with a variance-covariance matrix and included "paper" nested in "species" as a nested random effect in these models. We also used year as a random effect to control its potential effect on migration timing.

We used Bayesian linear mixed models to determine the effect of body mass on the durations of the four periods (breeding, non-breeding, spring and autumn migration) of migratory birds using the *brms* package[52]. Breeding latitude, non-breeding latitude and migration distance were also included in the models as explanatory variables; phylogenetic relationship among species (as mentioned above), year of tracking, paper ID and species ID were included in the models as random effects.

All the explanatory variables were scaled (mean zero, unit variance) prior to analysis. We did not detect strong multicollinearity among the variables in the analysis (all variance inflation factors (VIFs) < 2). We ran four Markov chains with 10,000 iterations and a burn-in of the first 1000 iterations per chain to contribute to summarizing the posterior distribution of estimation. Using the standard priors, the model converged with Rubin–Gelman statistics (<1.1)

according to Gelman and Rubin's diagnostic[62] model assessment was performed using approximate leave-one-out cross-validation (LOOIC) in the *loo* package of R[63]. We also used the *pp_check* function to assess the validity of Bayesian SEM model with posterior predictive checks in *brms* package[52] (Supplementary Fig. 16). To confirm causal pathways, we performed piecewise structural equation modelling using *piecewiseSEM* package before performing Bayesian SEM model[53]. This ensured results robustness in Bayesian SEM. The coefficient estimates of population-level effects with a 95% credible interval did not overlap zero.

Sex can influence migration timing[64], the selection of non-breeding sites[65] and, thus, migration distance due to differences in the roles in reproductive activities and resource competition between the males and the females. Hence, we further used SEM to investigate whether migration timing varied between sexes across the annual cycle using a dataset including data from 1077 individuals, accounting for 63% of the full dataset, after removing records with "unknown sex".

### Reporting summary

Further information on research design is available in the Nature Portfolio Reporting Summary linked to this article.

## Data availability

The data in this paper come from published papers, which are listed in Supplementary Data 1. All migration timing data used for the analyses are available in an online repository at Figshre[66]. The body mass data is available from the Handbook of Avian Body Masses[50]. Source data are provided with this paper.

## Code availability

The data analysis codes used in this study are all open-source software, which is available in an online repository at Figshare[66].

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

## Acknowledgements
This study was financially supported by the National Key Research and Development Program of China (grant number 2023YFF1304504) and the National Natural Science Foundation of China (grant numbers 31830089 and 31772467). We thank Chenchen Feng, Shaoping Zang, and Qianyan Zhou for their assistance in checking and correcting data. We thank all researchers contributing to bird tracking and sharing tracking data.

## Author contributions
Z.M. and X.W. conceived the project. X.W., W.C. and C.C. collected the migration dataset. X.W. performed the analyses. X.W., M.S., A.D., W.C., C.C., J.L. and Z.M. contributed to the first draft and the revisions.

## Competing interests
The authors declare no competing interests.
