## [Peer Review File · Nature Communications]

Macro-scale relationship between body mass and timing of bird migrationREVIEWER COMMENTS

Reviewer #1 (Remarks to the Author):

I enjoyed reading the ms which compiled the largest dataset on bird tracking studies. The authors used body mass, latitude and migration distance as key explanatory variables to explain the variability in migration timings and durations in a set of 186 species. This study has the potential to be a significant contribution to the field of animal migration.

In general, the introduction is very weak with too many hypothetical constructs for which I do not see any support in cited literature or in theory.

One common mistake that many researchers from northern latitudes make is that they refer to the long non-breeding period during Boreal winter as wintering period and the locations as wintering grounds. For anybody who lives in the Southern Hemisphere this must be odd. Similarly, I do not think that birds staying near equator are wintering there. Please, throughout the ms change the term wintering to the term non-breeding period/sites or whatever else.

In the abstract and introduction, you use some terminology in a very free manner but there is some tradition for people in the field about the meaning of particular terms. Body mass is not an endogenous factor but a species' life history trait. Please, reword this carefully throughout the ms.

Just to show your inconsistency on body mass: L 18 You use the term endogenous driver, L 21 you use the term endogenous factor, L 53 you use internal factor for body mass.

When on L18 you use the term endogenous driver, every student of bird migration will think that your study will be about internal clocks.

The last sentence of the abstract is an empty overstretching of your findings. I suggest deleting it.

L58 Please provide references for your claim that larger species have longer breeding periods. I could argue that smaller species often have several broods per season and this can overrun the large-bodied species.

L59 Larger-bodied species have in total longer moult but they often split it into seasons or years. Thus I do not think your link makes here a logical argument.

L60 Migration speed is widely understood as the distance covered between breeding and non-breeding grounds divided by total travel time (thus including stopovers). Just to make sure we are referring to the same variable as it often gets confused. But in general the actual flight speed scales positively with body mass.

L6264 You claim that breeding, moulting and migration do not overlap. This is a very coarse simplification. In fact many birds moult during breeding (e.g. many birds of prey), and many species start moulting while finishing their breeding cycle. There are plenty of references on that.

L64-65 You expect larger-bodied species to exhibit earlier migration. Could you please provide references on that? As far as I know, there is not much evidence for (e.g. <https://doi.org/10.1111/j.1365-2486.2009.01876.x>) that, but maybe I just overlooked some studies.

L65 Again, you need to better justify your prediction that larger species would migrate later due to time constraints. I could argue, again, that small-bodied species are often multiple-breeders and they extend their migration late into autumn. I simply do not believe this idea as I do not see a good theory of why it should be that way and I could list many large-bodied species that migrate early.

L69, L203 Change law to rule

L70 I think your idea that high latitudes are unfavourable for large species lacks support. Nestlings of many northern breeders are independent shortly after hatching and the parent can leave ahead the breeding grounds (e.g. waders).

L73-74 Yes, you are right about the role of body mass for potential cold tolerance and non-breeding latitudes, but your claim that these species stay at northern latitudes in order to shorten migration distance and the migration period is absolutely based on shaky grounds and you should provide better theoretical arguments for such claim.

L97 Please, use the term body mass throughout the ms. I do not see any rationale for why you switch to body size when this refers to a completely different biometric trait.

L104 You predict that larger species will not breed at higher latitudes. You cannot test for this prediction with your data. The scale of your study is as good as are your data and the essential problem is that tracking studies do not pick up study species randomly. Thus your set of 186 bird species is not a latitudinally representative sample of species. Researchers track species for various reasons. To assess this you would need to use the entire BirdLife dataset of species ranges and plot it against body masses. But this would be for another study.

L205-207 I think here you are in circular argumentation. Naturally, when large-bodied species spend the non-breeding period in Northern latitudes the migration distance to breeding sites must be short. That is all what is in it and no other speculations of why they have short distances are needed. Thus I suggest to drop your speculation about your statement that ...as it reduces the energy and time spent on migration.

L209 I disagree with your statement that smaller species have lower requirements for breeding and moulting periods. You use a very vague term of lower requirements. Please, be clear and give examples of what you mean, along with some species and case studies.

L248 Please rephrase this sentence as it is too strong in conclusion about endogenous factors. First of all, you use body mass, not size. Body mass is a trait, not an endogenous factor and please do not use plural form factors when you use only one variable – body mass.

L257 You give examples about the impacts of climatic change on species traits, and distribution. When there is ample evidence about shrinkage of body sizes, wouldn't that mean also, considering your results, that migration distances should increase as a consequence of climatic change? Can you elaborate on this? But as far as I can remember I know only about studies that report shrinkage in migration distances due to mild winters (white storks, blackcaps, some Parus species in Europe).

Regarding the results, in general, I find your findings very strong and I enjoyed reading this exciting ms. One thing that will be of interest to many readers will be the clade effects within birds. Please, make an overview of taxa covered in Table S1 by e.g. orders into another table. Now you use alphabetic ordering and it is not easy to have an overview of which taxa or families you use for your study. And within the most common groups, e.g. songbirds I would like to see the body mass and latitude effects. I am curious whether your patterns hold within taxonomic groups for which you have a reasonable number of available studies.

Methodologically, I think you should have used bird body masses directly from the studies where you extracted tracking data and not the averaged values from Dunning's book. Bear in mind that Bergmann's rule works intraspecifically, thus you can have considerable differences in mass between northern and southern populations of the same species.

One key issue is data availability. Although you state that you provided a full data availability statement in the ms, the fact is that you do not provide the data neither the code for your analyses.

As a consequence, I as a reviewer cannot check on your data extraction steps from published tracking studies and nobody can repeat your analyses. I think you should deposit all your extracted data into a data depository and provide a code for your analyses.

Reviewer #2 (Remarks to the Author):

This is an impressive piece of work that sets out to investigate how body size might influence patterns of phenology in migratory birds. The team have compiled an impressive data set and use a powerful phylogenetically controlled set of analyses to address a set of questions that arise from some of the classic textbook biogeographic patterns (i.e. the widely observed latitudinal gradients in the body sizes of animals). This is without doubt the most taxonomically broad analysis of this nature, it is global in extent, and this is the greatest achievement of the work. I guess it would be easy to point out the some of the results are confirmatory (e.g. males arriving first and the link between departure and arrival dates), but the scale is the key here and I do think the authors could make more of this.

I do think the results will be of broad interest to scientists (avian biologists, migration biologists and biogeographers) and it will likely generate interest among the public given the widespread fascination with the topic.

I have very few substantive comments (other referees will be better placed to evaluate the analytical approach more robustly), however I do have a number of minor thoughts in addition to those above:

1. The authors overstate the lack of work on endogenous drivers of phenology. There is a lot of work out there that has looked at fat stores and body condition. I think some consideration of these is needed. As outlined above the real progress here is the scale of the analysis.
2. There are also studies showing the endogenous nature of migratory timing (e.g. clocks) and these are worth mentioning in the preamble
3. The summary of broad migration patterns at the start of the results section is not really that informative and could go in the SOM.
4. I would be good see more detail on the body size and equator analyses, as I suspect these might be driven by the dominance of migratory waterfowl at the more northerly latitudes.

RESPONSE TO REVIEWERS' COMMENTS

Reviewer #1 (Remarks to the Author):

I enjoyed reading the ms which compiled the largest dataset on bird tracking studies. The authors used body mass, latitude and migration distance as key explanatory variables to explain the variability in migration timings and durations in a set of 186 species. This study has the potential to be a significant contribution to the field of animal migration.

Response: Thank you for your positive and encouraging comment on our study. We have carefully revised the MS as suggested. Please see below for our point-by-point responses (begin with "Response:" and in bold).

In general, the introduction is very weak with too many hypothetical constructs for which I do not see any support in cited literature or in theory.

Response: Thank you for pointing out the weaknesses in the Introduction. We have carefully edited the Introduction section, rephrased hypotheses and cited suitable literature to support our statements. Please see our response to the comments below for the details. We hope you will find the revision is fine.

One common mistake that many researchers from northern latitudes make is that they refer to the long non-breeding period during Boreal winter as wintering period and the locations as wintering grounds. For anybody who lives in the Southern Hemisphere this must be odd. Similarly, I do not think that birds staying near equator are wintering there. Please, throughout the ms change the term wintering to the term non-breeding period/sites or whatever else.

Response: We agree and have changed the "wintering periods/sites" to "non-breeding periods/sites" throughout the MS in the revision.

In the abstract and introduction, you use some terminology in a very free manner but there is some tradition for people in the field about the meaning of particular terms. Body mass is not an endogenous factor but a species' life history trait. Please, reword this carefully throughout the ms. Just to show your inconsistency on body mass: L 18 You use the term endogenous driver, L 21 you use the term endogenous factor, L 53 you use internal factor for body mass. When on L18 you use the term endogenous driver, every student of bird migration will think that your study will be about internal clocks.

Response: We apologize for the confusion and have revised the description as suggested. In the revision, body mass was used as a species' biological trait throughout the MS. "

The last sentence of the abstract is an empty overstretching of your findings. I suggest deleting it.

Response: Done as suggested.

L58 Please provide references for your claim that larger species have longer breeding periods. I could argue that smaller species often have several broods per season and this can overrun the large-bodied species.

Response: Thank you for pointing this out. We agree with you. To clarify the issue, we show that larger species have longer breeding periods for one brood. We cited the modelling results of Hedenström (2006), who used a combination of allometric relationships and theoretical deduction to show that larger species tend to have longer breeding seasons. We revised the sentence as: “An analysis of the scaling of annual cycle in birds suggests that larger species spend longer periods breeding one brood, which is largely driven by longer incubation periods due to larger eggs¹, and longer time to grow into adult size².” Please see Lines 58-60.

We agree that smaller species can have several broods per season, which will extend their breeding period. As a consequence, we revised the hypothesis in Lines 66-70 as: “Although the duration of the breeding period per brood for larger species is generally longer, smaller species may produce multiple broods during the breeding season when the timing is suitable. Therefore, there might be no consistent effect of body mass on the departure date of migratory birds from their breeding sites”.

L59 Larger-bodied species have in total longer moult but they often split it into seasons or years. Thus I do not think your link makes here a logical argument.

Response: You are right. We removed this hypothesis from the text.

L60 Migration speed is widely understood as the distance covered between breeding and non-breeding grounds divided by total travel time (thus including stopovers). Just to make sure we are referring to the same variable as it often gets confused. But in general the actual flight speed scales positively with body mass.

Response: We agree. To clarify the issue, we revised the sentence to “...migration speed (i.e., migration distance divided by the total travel time) decreases with increasing body mass during powered flapping flights....” Please see the Lines 60-62.

L62-64 You claim that breeding, moulting and migration do not overlap. This is a very coarse simplification. In fact many birds moult during breeding (e.g. many birds of prey), and many species start moulting while finishing their breeding cycle. There are plenty of references on that.

Response: We agree and have removed the content related to moulting from the text in the revision. Therefore, the revised assumption is that breeding and migration are separated in time, which is a more reasonable assumption. Please see Lines 64-65.

L64-65 You expect larger-bodied species to exhibit earlier migration. Could you please provide references on that? As far as I know, there is not much evidence for (e.g. <https://doi.org/10.1111/j.1365-2486.2009.01876.x>) that, but maybe I just overlooked some studies.

Response: There is evidence in shorebirds that larger species start migration earlier in spring and we added the reference (Zhao et al. 2018, *Journal of Avian Biology*, 49, e01570) here in the revision. We revised the sentence as: “...larger species exhibit earlier migration timing in spring, which has been observed in some shorebird species³.” Please see Lines 65-66.

We agree that the expectation that larger species depart later from the breeding grounds might be not justified. Following your comment below, we revised the sentence as: “Although the duration of breeding period per brood for larger species is generally longer, smaller species may produce multiple broods during the breeding season when the timing is suitable. Therefore, there might be no consistent effect of body mass on the departure date of migratory birds from their breeding sites.” Please see Line 66-70.

L65 Again, you need to better justify your prediction that larger species would migrate later due to time constraints. I could argue, again, that small-bodied species are often multiple-breeders and they extend their migration late into autumn. I simply do not believe this idea as I do not see a good theory of why it should be that way and I could list many large-bodied species that migrate early.

Response: We agree and have revised the prediction. Please see our previous response.

L69, L203 Change law to rule

Response: Done as suggested.

L70 I think your idea that high latitudes are unfavourable for large species lacks support. Nestlings of many northern breeders are independent shortly after hatching and the parent can leave ahead the breeding grounds (e.g. waders).

Response: We revised the sentence to “However, breeding seasons are relatively short at high latitudes, which may limit the ability of larger species to breed there as they tend to have a longer time requirement for the breeding period”. Please see the Lines 75-77.

L73-74 Yes, you are right about the role of body mass for potential cold tolerance and non-breeding latitudes, but your claim that these species stay at northern latitudes in order to shorten migration distance and the migration period is absolutely based on shaky grounds and you should provide better theoretical arguments for such claim.

Response: Sorry for the confusing description. We revised the sentence as: “We therefore predict that larger species tend to breed at lower latitudes than smaller species, but that they spend the non-breeding season at relatively high latitudes compared to smaller species due to their higher tolerance to colder climate, which would result in a shorter migration distance and migration period.”. Please see Lines 77-80.

L97 Please, use the term body mass throughout the ms. I do not see any rationale for why you switch to body size when this refers to a completely different biometric trait.

Response: Done as suggested. We used the term body mass throughout the MS in the revision.

L104 You predict that larger species will not breed at higher latitudes. You cannot test for this prediction with your data. The scale of your study is as good as are your data and the essential problem is that tracking studies do not pick up study species randomly. Thus your set of 186 bird species is not a latitudinally representative sample of species. Researchers track species

for various reasons. To assess this you would need to use the entire BirdLife dataset of species ranges and plot it against body masses. But this would be for another study.

Response: Sorry for the inaccurate description. We revised the sentence to “...larger species tend to breed at lower latitudes...”. We also exhibited the body mass distribution of the 186 tracked species compared with that of all migratory species in the world in the revision to show the representativeness of the samples. Please see Line 77 and Supplementary Fig. 13 in the revision.

L205-207 I think here you are in circular argumentation. Naturally, when large-bodied species spend the non-breeding period in Northern latitudes the migration distance to breeding sites must be short. That is all what is in it and no other speculations of why they have short distances are needed. Thus I suggest to drop your speculation about your statement that ...as it reduces the energy and time spent on migration.

Response: Done as suggested. Please see Lines 223-224.

L209 I disagree with your statement that smaller species have lower requirements for breeding and moulting periods. You use a very vague term of lower requirements. Please, be clear and give examples of what you mean, along with some species and case studies.

Response: Sorry for the confusing description. As mentioned earlier, we cited Hedenström (2006) to show that larger species tend to have longer breeding seasons for one brood, and we removed description on moulting in the revision. We also specify the lower time and energy requirement for smaller species per brood. We revised this sentence to “...long-distance migratory birds, especially those wintering in the Southern Hemisphere, tend to be smaller species (Fig. 2, Supplementary Fig. 5b), which have relatively low time and energy requirements for breeding per brood compared with larger species, potentially allowing them to invest more time and energy in migration activities”. Please see Lines 221-224.

L248 Please rephrase this sentence as it is too strong in conclusion about endogenous factors. First of all, you use body mass, not size. Body mass is a trait, not an endogenous factor and please do not use plural form factors when you use only one variable – body mass.

Response: Done as suggested. We revised the sentence as: “This study illustrates how environmental factors (latitude and distance), biological trait (body mass), and their interaction can influence key timings in the annual cycle of migratory birds”. Please see lines 267-268.

L257 You give examples about the impacts of climatic change on species traits, and distribution. When there is ample evidence about shrinkage of body sizes, wouldn't that mean also, considering your results, that migration distances should increase as a consequence of climatic change? Can you elaborate on this? But as far as I can remember I know only about studies that report shrinkage in migration distances due to mild winters (white storks, blackcaps, some Parus species in Europe).

Response: According to our results, shrinkage of body sizes provides the ability to increase migration distance. Meanwhile, other factors (such as available habitats and

resources, interspecies competition, etc.) can also affect migration distance increase. We agree that shrinkage in migration distance (northward movement of non-breeding grounds) is common in migratory species due to mild winter. It might be also possible that migration distance increases in the Southern Hemisphere if birds with shrinkage body size can obtain better resources. We revised the sentences to: “In response to global warming, many migratory birds advance migration timing in spring. In addition, some species are showing shrinking body sizes, shifts in their breeding and non-breeding ranges and corresponding changes in migration distances. Our results suggest that all these changes can also affect migration timing both directly and indirectly...”. Please see Lines 272-278.

Regarding the results, in general, I find your findings very strong and I enjoyed reading this exciting ms. One thing that will be of interest to many readers will be the clade effects within birds. Please, make an overview of taxa covered in Table S1 by e.g. orders into another table. Now you use alphabetic ordering and it is not easy to have an overview of which taxa or families you use for your study. And within the most common groups, e.g. songbirds I would like to see the body mass and latitude effects. I am curious whether your patterns hold within taxonomic groups for which you have a reasonable number of available studies.

Response: We revised Table S1 as suggested. We reordered the species according to the phylogenetic relationship according to the maximum clade credibility tree derived from BirdTree.org, and we also added two columns of order and family in Supplementary Table 1 and Table 2.

As suggested, we analyzed the impacts of body mass on migration timing in Passeriformes with data from over 50 species. We found that the migration timing of Passeriformes was strongly affected by breeding latitude but not body mass. Please see the Supplementary Figure 8. We also added the related contents in the Results section: “Further analysis on Passeriformes (53 tracked species) indicated that breeding latitude but not body mass significantly affected annual migration timing (Supplementary Fig. 8).” Please see Lines 162-163. We also discussed the result in the revision: “We found that the migration timing of Passeriformes was strongly affected by breeding latitude but not body mass. This might be due to the smaller body mass of Passeriformes (< 90 g for the 53 species in this study) makes them sensitive to periodical environmental changes, and thus annual migration timing is endogenously controlled⁴. This also suggests the impacts of body mass on migration timing mainly occur among taxonomic groups.” Please see Lines 243-248.

Methodologically, I think you should have used bird body masses directly from the studies where you extracted tracking data and not the averaged values from Dunning’s book. Bear in mind that Bergmann’s rule works intraspecifically, thus you can have considerable differences in mass between northern and southern populations of the same species.

Response: Body mass of many migratory birds largely varies among periods due to fuel deposition within a year. For example, some long-distance migratory birds nearly double their body mass before non-stop migratory flight of several or tens of thousands of kilometers⁵. Therefore, we used the minimum body mass of females or males from

Dunning's book can be representative of the lean body mass of birds. Moreover, some studies did not show the body mass of tagged birds at capture, making the data from the studies unavailable. In the data analysis, we defined species as a random effect and focused on detecting the effect of body mass on timing among but not within species.

One key issue is data availability. Although you state that you provided a full data availability statement in the ms, the fact is that you do not provide the data neither the code for your analyses. As a consequence, I as a reviewer cannot check on your data extraction steps from published tracking studies and nobody can repeat your analyses. I think you should deposit all your extracted data into a data depository and provide a code for your analyses.

Response: We have thought that the data in this MS are not ours but come from published papers of other researchers, thus we did not provide the full data during the earlier submission. As suggested, we have provided the full data (showing the data sources) and code for data analysis in this revision.

Reviewer #2 (Remarks to the Author):

This is an impressive piece of work that sets out to investigate how body size might influence patterns of phenology in migratory birds. The team have compiled an impressive data set and use a powerful phylogenetically controlled set of analyses to address a set of questions that arise from some of the classic textbook biogeographic patterns (i.e. the widely observed latitudinal gradients in the body sizes of animals). This is without doubt the most taxonomically broad analysis of this nature, it is global in extent, and this is the greatest achievement of the work. I guess it would be easy to point out the some of the results are confirmatory (e.g. males arriving first and the link between departure and arrival dates), but the scale is the key here and I do think the authors could make more of this.

I do think the results will be of broad interest to scientists (avian biologists, migration biologists and biogeographers) and it will likely generate interest among the public given the widespread fascination with the topic.

Response: Thank you for your positive comments on our study.

I have very few substantive comments (other referees will be better placed to evaluate the analytical approach more robustly), however I do have a number of minor thoughts in addition to those above:

1. The authors overstate the lack of work on endogenous drivers of phenology. There is a lot of work out there that has looked at fat stores and body condition. I think some consideration of these is needed. As outlined above the real progress here is the scale of the analysis.

Response: We apologize for the confusion in our description. In this study, we want to emphasize the effect of body mass on migration timing. Another reviewer also pointed out the issue. In the revision, we used "biological trait" instead throughout the MS.

2. There are also studies showing the endogenous nature of migratory timing (e.g. clocks) and these are worth mentioning in the preamble

Responses: We revised the sentence in the Introduction as: “The annual cycle of migratory birds is composed of a series of events and biological activities that are strictly timed, generally by internal biological clock⁶, to match...”. Please see Lines 34-37 in the revision.

3. The summary of broad migration patterns at the start of the results section is not really that informative and could go in the SOM.

Responses: We understand that the summary of migration patterns is not the core of our results, but we want to show an overall migration timing to the readers. We hope it’s OK to leave the summary in the first paragraph of the Results section.

4. I would be good see more detail on the body size and equator analyses, as I suspect these might be driven by the dominance of migratory waterfowl at the more northerly latitudes.

Responses: Migratory waterfowl including geese and swans (e.g., *Anser cygnoides* and *Cygnus olor*) really stay at more northerly latitudes during non-breeding season. Other large-sized species, such as some cranes (e.g., *Grus leucogeranus*), raptors (e.g., *Haliaeetus leucocephalus*), storks (e.g., *Ciconia ciconia*), and loons (e.g., *Gavia immer*), also stay in the North Hemisphere during the non-breeding season. We revised the sentence as: “... the average latitudes of non-breeding sites for each species weighing more than 1.1 kg (36 species, including some geese, cranes, storks, and eagles, etc.) were all located north of the equator (Supplementary Fig. 5b)”. Please see Lines 146-148.

References

- 1 Hedenström, A. Scaling of migration and the annual cycle of birds. *Ardea* **94**, 399-408 (2006).
- 2 Ricklefs, R. E. Patterns of growth in birds. *Ibis* **110**, 419-451 (1968).
- 3 Zhao, M. *et al.* Body size shapes inter-specific migratory behaviour: evidence from individual tracks of long-distance migratory shorebirds. *J. Avian Biol.* **49**, e01570 (2018).
- 4 Briedis, M. *et al.* Breeding latitude leads to different temporal but not spatial organization of the annual cycle in a long-distance migrant. *J. Avian Biol.* **47**, 743-748 (2016).
- 5 Lindström, Å. & Piersma, T. Mass changes in migrating birds: the evidence for fat and protein storage re-examined. *Ibis* **135**, 70-78 (1993).
- 6 Åkesson, S. *et al.* Timing avian long-distance migration: from internal clock mechanisms to global flights. *Philos. Trans. R. Soc. B: Biol.* **372** (2017).

REVIEWER COMMENTS

Reviewer #1 (Remarks to the Author):

I would like to thank the authors for their substantial revisions and for very careful and critical assessments of my previous comments on the ms. I think the revised text is now much better, and this work will be of great reference to many of those who study animal migration. You have done really a great job.

I am happy with the ms, but I do have one more and last critical issue to consider. It comes to your hypothesis no. 1 (L109–110): that the body mass of migratory birds affects latitudinal distribution of their breeding and non-breeding sites.

While the second part of the hypothesis about non-breeding sites is absolutely OK, with the second part, given the nature of your data, I have a methodological problem. The problem is not with your idea but with your data. I think the way the data were collected in the field studies does not allow you to test the first part of the hypothesis on breeding sites. To be able to answer this question you would need a dataset where breeding sites are sampled randomly. However, this is not the case with the compiled dataset. Most tracking studies were conducted in such a way that the researchers caught the birds at breeding sites. The breeding site is usually a single plot/dot in your analysis. Thus, the northern locations of your data points are nonrandomly sampled. In contrast, the same birds caught at the single breeding site will later, in Boreal winter, disperse across a wide range of latitudes during the non-breeding period. Vice versa, when somebody traps the birds at the non-breeding location (e.g. in Africa or S America) that site in your analysis will have almost the same latitude for all individuals while the breeding latitudes in the North will spread across a huge area. Do you get my point?

The only way how to tackle this issue is either to drop the hypothesis that body mass affects latitudes of breeding sites or to pick up studies that trapped birds at non-breeding sites and use this subset for assessment of your hypothesis.

This touches also the results on L142–143. The fact that you did not find an effect of body mass on breeding latitude merely reflects the constraint in the sampling of studies = study sites picked up by researchers are your data points.

And again I suggest that the discussion on L200–208 reflects on this = is deleted/rewritten.

L 230, 237 Sorry for being again so fussy about terms, but it does matter. Here you use the term breeding period, but in your study, you did not assess the breeding period per se – in your case, it also includes the post-breeding moult period. I would suggest rephrasing it e.g.: ... larger birds spent longer time at the breeding sites than smaller birds.

Reviewer #2 (Remarks to the Author):

I am happy with the corrections made.. all the points I have raised have been clarified. I have one final point. I think it is good that the team have moved away from using 'endogenous' when referring to the distinction between traits... however, I feel 'biological traits' is a bit too vague as it encompasses many of the the aspects they are trying to distinguish. Maybe use 'intrinsic biological traits' to make it clear where the difference lies.

Reviewer #3 (Remarks to the Author):

Reviewer's evaluation for Wang et al. 2023

The ms provides a highly interesting research question and applies cutting-edge statistical methods to solve it.

Considering your working hypotheses, i) I accept your expectation that to maximize fitness larger species exhibit earlier migration timing in spring; ii) you predicted that body mass affects the latitudinal distribution of species which thus can indirectly affect migration timing following Bergmann's rule, which is fully acceptable; iii) I accept your expectation that breeding seasons are relatively short at high latitudes, which may limit the ability of larger species to breed there as they tend to have a higher time requirement for the breeding period; iv) I accept the prediction that larger species tend to breed at lower latitudes than smaller species, but that they spend the non-breeding season at relatively high latitudes compared to smaller species due to their higher tolerance to colder climate; v) I accept your reasoning that body mass is related to the latitude of breeding and non-breeding sites following Bergmann's rule; body mass also impacts the cost of migration; consequently, body mass can directly and indirectly affect migration distance.

Methods

How representative is your set of 186 bird species for the complete species pool which breed in the Northern Hemisphere?

You considered the average latitudes of non-breeding sites for each species weighing more than 1.1 kg; how did you choose this limit?

I agree with your approach that for individuals which stayed at more than one non-breeding site within one season, you used the first non-breeding site they visited and the date of arrival at that site as the non-breeding site and the arrival date for the autumn migration, respectively, and you used the last non-breeding site they visited and the date of departure from that site as the non-breeding site and the departure date from the non-breeding site in the spring migration.

I agree that you defined the distance between the breeding site and the first non-breeding site as the migration distance in autumn and the distance between the last non-breeding site and the breeding site as the migration distance in spring.

I agree with your approach that if sex was not reported in the study, you used the average of the minimum body masses of both sexes.

I agree that birds can reset migration timing during the non-breeding period, so the migration timing at the earlier annual cycle has no carry-over effects on the migration timing at the following annual cycle.

I fully agree with the inclusion of the flight mode in the SEMs.

I fully agree with the approach of controlling for phylogenetic dependence in the models, by i) randomly sampling 1000 pseudo-posterior distributions; ii) constructing the maximum clade credibility tree using common ancestor heights with TreeAnnotator software from the BEAST package; and iii) calculating the phylogenetic similarity at the species level with a variance-covariance matrix.

Although I agree with your approach of model development by including breeding latitude, non-breeding latitude and migration distance as explanatory variables, and including phylogenetic relationship among species year of tracking, paper ID and species ID in the models as random effects, please show that including phylogenetic relationship on the species level in the form of a variance-covariance matrix and also including species as random effect term results in no mathematical violation of the assumptions of the statistical models.

Based on the above statements, I fully agree with the structure of the Bayesian SEM component models. Using the R code provided by the authors, I was able to reproduce the results and thus

accept these.

Dr. Zsolt Végvári

RESPONSE TO REVIEWERS' COMMENTS

Reviewer #1 (Remarks to the Author):

I would like to thank the authors for their substantial revisions and for very careful and critical assessments of my previous comments on the ms. I think the revised text is now much better, and this work will be of great reference to many of those who study animal migration. You have done really a great job.

Response: Thank you very much for recognizing our efforts in revising the manuscript. Your constructive comments have been instrumental in enhancing the quality of our work. We are pleased to hear that the revised text meets your expectations and hope it will serve as a valuable reference in the field of animal migration studies. Please see below for our point-by-point responses (begin with “Response:” and in bold).

I am happy with the ms, but I do have one more and last critical issue to consider. It comes to your hypothesis no. 1 (L109–110): that the body mass of migratory birds affects latitudinal distribution of their breeding and non-breeding sites.

While the second part of the hypothesis about non-breeding sites is absolutely OK, with the second part, given the nature of your data, I have a methodological problem. The problem is not with your idea but with your data. I think the way the data were collected in the field studies does not allow you to test the first part of the hypothesis on breeding sites. To be able to answer this question you would need a dataset where breeding sites are sampled randomly. However, this is not the case with the compiled dataset. Most tracking studies were conducted in such a way that the researchers caught the birds at breeding sites. The breeding site is usually a single plot/dot in your analysis. Thus, the northern locations of your data points are nonrandomly sampled. In contrast, the same birds caught at the single breeding site will later, in Boreal winter, disperse across a wide range of latitudes during the non-breeding period. Vice versa, when somebody traps the birds at the non-breeding location (e.g. in Africa or S America) that site in your analysis will have almost the same latitude for all individuals while the breeding latitudes in the North will spread across a huge area. Do you get my point?

The only way how to tackle this issue is either to drop the hypothesis that body mass affects latitudes of breeding sites or to pick up studies that trapped birds at non-breeding sites and use this subset for assessment of your hypothesis.

Response: Thank you for pointing out this issue. Although I think with a sampling size of 186 species, nonrandom sampling is not a serious problem, I agree with your view. To solve the issue of nonrandom sampling, we followed your suggestion with further assessing the relationship between body mass and breeding latitude by using the combined data of birds captured at stopover (7 species) and non-breeding sites (45 species). We also assessed the relationship between body mass and non-breeding latitude by using the data of birds captured at breeding sites (154 species).

We found the results were consistent with earlier: There was significant correlation between body mass and non-breeding latitudes, with larger species tending to stay at higher latitudes in non-breeding season (95% CI [0.32, 0.79]); body mass did not exhibit a significant latitudinal gradient in the breeding season (95% CI [-0.54, 0.23]). We

added the results in the revision. Please see Lines 140-146 in the revision and Supplementary Fig. 5.

In the “Methods” section, we also revised the sentences as: “We extracted the following complete data from each selected paper: species and/or subspecies name, ..., capture site and year when tracking was conducted...” Please see Lines 426-430. We also added “To address the issue of nonrandom sampling in analyzing the relationship between body mass and breeding/non-breeding latitudes, we further assessed the relationship between body mass and breeding latitude by using the combined data of birds captured at stopover and non-breeding sites. We also assessed the relationship between body mass and non-breeding latitude by using the data of birds captured at breeding sites.” Please see Lines 510-515.

For the convenience of checking the data, we added one column “capture site” in our dataset in the revision. We also included the relevant analysis code in this revision. Please see the raw data and code in Figshare.

This touches also the results on L142–143. The fact that you did not find an effect of body mass on breeding latitude merely reflects the constraint in the sampling of studies = study sites picked up by researchers are your data points.

Response: We revised the sentences and added the results as: “SEM analyses demonstrated a significant correlation between body mass and non-breeding latitudes of the tracked birds, with larger species tending to winter at higher latitudes (95% CI [0.33, 0.78] with full dataset, Figs. 2 and 3; 95% CI [0.32, 0.79] with data of birds captured at breeding sites, Supplementary Fig. 5b). However, body mass did not exhibit a significant latitudinal gradient in the breeding season (95% CI [-0.12, 0.35] with full dataset, Fig. 2 and Supplementary Fig. 6a; 95% CI [-0.54, 0.23] with data of birds captured at non-breeding and stopover sites, Supplementary Figs. 5a and 6b).” Please see Lines 140-146, and Supplementary Fig. 5.

And again I suggest that the discussion on L200–208 reflects on this = is deleted/rewritten.

Response: According to the supplied analysis, we revised the sentence as: “...body mass does not exhibit a significant latitudinal gradient in migratory birds during their breeding season, according to data of all the birds or data of birds captured outside the breeding sites.” Please see Lines 203-205.

L 230, 237 Sorry for being again so fussy about terms, but it does matter. Here you use the term breeding period, but in your study, you did not assess the breeding period per se – in your case, it also includes the post-breeding moult period. I would suggest rephrasing it e.g.: ... larger birds spent longer time at the breeding sites than smaller birds.

Response: Sorry for the confusing description. Done as suggested. We revised the sentence as: “...larger birds spent longer time at the breeding sites than smaller birds...”. (Please see Lines 234-235). “The longer duration in the breeding sites of larger birds was compensated by a shorter duration in the non-breeding site...” (Please see Lines 241-242).

Reviewer #2 (Remarks to the Author):

I am happy with the corrections made. all the points I have raised have been clarified. I have one final point. I think it is good that the team have moved away from using 'endogenous' when referring to the distinction between traits... however, I feel 'biological traits' is a bit too vague as it encompasses many of the the aspects they are trying to distinguish. Maybe use 'intrinsic biological traits' to make it clear where the difference lies.

Response: Thank you for your suggestion. We agree and have changed the “biological traits” to “intrinsic biological traits” throughout the MS in the revision.

Reviewer #3 (Remarks to the Author):

Reviewer’s evaluation for Wang et al. 2023

The ms provides a highly interesting research question and applies cutting-edge statistical methods to solve it.

Response: Thank you for your positive comments on our study.

Considering your working, hypotheses, i) I accept your expectation that to maximize fitness larger species exhibit earlier migration timing in spring; ii) you predicted that body mass affects the latitudinal distribution of species which thus can indirectly affect migration timing following Bergmann's rule, which is fully acceptable; iii) I accept your expectation that breeding seasons are relatively short at high latitudes, which may limit the ability of larger species to breed there as they tend to have a higher time requirement for the breeding period; iv) I accept the prediction that larger species tend to breed at lower latitudes than smaller species, but that they spend the non-breeding season at relatively high latitudes compared to smaller species due to their higher tolerance to colder climate; v) I accept your reasoning that body mass is related to the latitude of breeding and non-breeding sites following Bergmann's rule; body mass also impacts the cost of migration; consequently, body mass can directly and indirectly affect migration distance.

Response: Thank you for accepting our hypotheses.

Methods

How representative is your set of 186 bird species for the complete species pool which breed in the Northern Hemisphere?

Response: We added the sentence: “...The 186 species were 12.0% of the total 1550 migratory species breeding in the Northern Hemisphere¹.” Please see Lines 446-447.

You considered the average latitudes of non-breeding sites for each species weighing more than 1.1 kg; how did you choose this limit?

Response: No special reason, we checked the data and found that all the heavier species were distributed in the Northern Hemisphere in non-breeding season, thus we exhibited the limit of 1.1 kg.

I agree with your approach that for individuals which stayed at more than one non-breeding site within one season, you used the first non-breeding site they visited and the date of arrival at that site as the non-breeding site and the arrival date for the autumn migration, respectively, and you used the last non-breeding site they visited and the date of departure from that site as the non-breeding site and the departure date from the non-breeding site in the spring migration.

Response: Thank you for your agreement.

I agree that you defined the distance between the breeding site and the first non-breeding site as the migration distance in autumn and the distance between the last non-breeding site and the breeding site as the migration distance in spring.

Response: Thank you for your agreement.

I agree with your approach that if sex was not reported in the study, you used the average of the minimum body masses of both sexes.

Response: Thank you for your agreement.

I agree that birds can reset migration timing during the non-breeding period, so the migration timing at the earlier annual cycle has no carry-over effects on the migration timing at the following annual cycle.

Response: Thank you for your agreement.

I fully agree with the inclusion of the flight mode in the SEMs.

Response: Thank you for your agreement.

I fully agree with the approach of controlling for phylogenetic dependence in the models, by i) randomly sampling 1000 pseudo-posterior distributions; ii) constructing the maximum clade credibility tree using common ancestor heights with TreeAnnotator software from the BEAST package; and iii) calculating the phylogenetic similarity at the species level with a variance-covariance matrix.

Response: Thank you for your agreement.

Although I agree with your approach of model development by including breeding latitude, non-breeding latitude and migration distance as explanatory variables, and including phylogenetic relationship among species year of tracking, paper ID and species ID in the models as random effects, please show that including phylogenetic relationship on the species level in the form of a variance-covariance matrix and also including species as random effect term results in no mathematical violation of the assumptions of the statistical models.

Response: Done as suggested. We showed the standard deviation (SD) of all random effects including phylogenetic relationship and species in Bayesian SEM. All the SDs of random effects were different from zero. Please see Supplementary Fig.10. In the revision,

we indicated that “...We also used the *pp_check* function to assess the validity of Bayesian SEM model with posterior predictive checks in *brms* package (Supplementary Fig. 16).” Please see Lines 545-547 and Supplementary Fig. 16.

Based on the above statements, I fully agree with the structure of the Bayesian SEM component models. Using the R code provided by the authors, I was able to reproduce the results and thus accept these.

Response: Thank you for your comments.

References

- 1 Tobias, J. A. *et al.* AVONET: morphological, ecological and geographical data for all birds. *Ecol. Lett.* **25**, 581-597 (2022).